# The Dissemination Strategy of an Urban Smart Medical Tourism Image by Big Data Analysis Technology

**DOI:** 10.3390/ijerph192215330

**Published:** 2022-11-20

**Authors:** Zijian Zhao, Zhongwei Wang, Javier Garcia-Campayo, Hector Monzales Perez

**Affiliations:** 1Department of Performing Arts and Culture, The Catholic University of Korea, Bucheon-si 14662, Republic of Korea; 2School of Journalism and Communication, University of Chinese Academy of Social Sciences, Beijing 102488, China; 3School of Communication and Film, Hong Kong Baptist University, Hong Kong 999077, China; 4College of Educations, Arts and Sciences, Lyceum of the Philippines University-Batangas, Batangas 4200, Philippines; 5School of Medicine, University of Zaragoza, 50009 Zaragoza, Spain; jgarcamp@unizar.es; 6Management School, The University of Sheffield, Sheffield S10 2TN, UK; 19448767@life.hkbu.edu.hk; 7Hospital Universitario Miguel Servet, 50009 Zaragoza, Spain; 8Republic of the Philippines Professional Regulation Commission, Manila 1008, Philippines; hector.perez@prc.gov.ph

**Keywords:** big data, city, smart medical treatment, tourism image, communication strategy

## Abstract

The advanced level of medical care is closely related to the development and popularity of a city, and it will also drive the development of tourism. The smart urban medical system based on big data analysis technology can greatly facilitate people’s lives and increase the flow of people in the city, which is of great significance to the city’s tourism image dissemination and branding. The medical system, with eight layers of architecture including access, medical cloud service governance, the medical cloud service resource, the platform’s public service, the platform’s runtime service, infrastructure, and the overall security and monitoring system of the platform, is designed based on big data analysis technology. Chengdu city is taken as an example based on big data analysis technology to position the dissemination of an urban tourism image. Quantitative analysis and questionnaire methods are used to study the effect of urban smart medical system measurement and tourism image communication positioning based on big data analysis technology. The results show that the smart medical cloud service platform of the urban smart medical system, as a public information service system, supports users in obtaining medical services through various terminal devices without geographical restrictions. The smart medical cloud realizes service aggregation and data sharing compared to the traditional isolated medical service system. Cloud computing has been used as the technical basis, making the scalability and reliability of the system have unprecedented improvements. This paper discusses how to effectively absorb, understand, and use tools in the big data environment, extract information from data, find effective information, make image communication activities accurate, reduce the cost, and improve the efficiency of city image communication. The research shows that big data analysis technology improves patients’ medical experience, improves medical efficiency, and alleviates urban medical resource allocation to a certain extent. This technology improves people’s satisfaction with the dissemination of urban tourism images, makes urban tourism image dissemination activities accurate, reduces the cost of urban tourism image dissemination, and improves the efficiency of urban tourism image dissemination. The combination of the two can provide a reference for developing urban smart medical care and disseminating a tourism image.

## 1. Introduction

Tourism image dissemination, to a certain extent, determines the popularity and reputation of urban tourism. It directly affects the development of tourism and determines the expansion of the urban tourism market. Tourism image dissemination is the key factor for a city’s tourism image to attract tourists. It also contributes to huge urban wealth and intangible urban soft assets owned by a city [1]. Currently, many cities have realized the significance and value of the urban tourism image for developing urban tourism resources. Many cities are making tourism development plans, investing funds, and designing and building their urban tourism image and brand. The development strategy of an urban tourism image has gradually become the core strategy of urban development. Although the value of an urban tourism image is significant, the dissemination of the tourism image needs to be gradual. This project is systematic [2]. Only by placing the strategic planning of urban tourism at the level of overall urban planning can the desired results be achieved. As the information society gradually develops to an advanced stage, the media and patterns of dissemination are also undergoing tremendous changes. The traditional mass media still exists and plays a role. The endless new media forms also occupy a lot of time and space in people’s daily lives. People live in a world surrounded by mobile Internet and big data. New technologies such as artificial intelligence, virtual reality/augmented reality, and digital media are affecting the ecological development of the media industry [3]. Especially in the era of big data, the audience is no longer simply expressed as people and their life status. Relevant smart devices, smart scenes, and all the data generated by the audience’s life trajectory are important for understanding users. From this perspective, the era of big data has, in fact, brought a new research perspective and approach to audience research [4]. As the soft power of a city, the reasonable development and utilization of a city image can create a favorable public opinion environment and development space for the city to promote the comprehensive competitiveness of the city. Throughout the current city image dissemination development, most cities’ image dissemination follows the following steps. The first is city image positioning. A city image communication strategy is developed. The second is to assess the effectiveness of city image dissemination. The third is to optimize the city image dissemination. The positioning of a city’s image relies on the city’s historical image, available resources, and expectations for the future. It is especially important to develop an effective city image dissemination strategy and integrate the means of dissemination media. It is a great challenge to combine the existing new and old media means to achieve the optimal combination of media, expand media reach, and increase the number of people covered by media [5].

Medical tourism is a new type of tourism service with the themes of medical treatment and nursing, illness and health, and rehabilitation and recuperation. At present, it has grown into one of the fastest-growing emerging industries in the world. It has maintained an average annual growth rate of 20% in recent years, far exceeding the overall growth rate of other tourism industries. Asian countries and regions such as Thailand, India, Singapore, South Korea, Jordan, Taiwan, Malaysia, and Japan, European countries such as Switzerland, Hungary, and Spain, countries in the Americas such as Brazil, Panama, and Costa Rica, and South Africa have become important medical tourism destinations [6]. In 2007, the medical tourism services revenue in Asia alone reached USD 34 billion, accounting for 12.7% of the global market share. Countries are attracting consumers around the world with the advantage of characteristic medical resources and have received huge foreign exchange earnings. The 2011 Annual Report on the Asian Economic Integration Process released by the 2011 Boao Forum for Asia said that Asia has become the world’s most potential medical tourism service market [7]. Experts expect China to be the next hot country for medical tourism. Traditional Chinese Medicine (TCM) has a strong appeal to foreigners. Medical tourism is a new type of industry that integrates the medical industry and tourism. It promotes regional economic development, and also drives the rapid development of tourism, medical services, local aviation, convention and exhibition, and other related industries. With the increase of local medical costs and the number of medical tourists, going to other places for medical treatment has become a new trend. Medical tourism can be divided into four types based on its origin characteristics. The first is a destination with a service advantage. The second is a destination with characteristic medical resources as an attraction. The third is a destination with a quality healthcare system as an attraction. The fourth is the attraction of high quality and low price [8]. At present, China’s medical tourism focuses on the development of TCM tourism resources and belongs to the type of destination with characteristic medical resources as the attraction point. While fully tapping the tourism resources of TCM in China, it is also important to improve the quality of services and improve the medical system. In the process of medical tourism development, many countries in the world have developed rapidly with low prices and good services [9]. For example, medical institutions in Thailand provide various medical treatments such as plastic surgery, beauty, and metamorphosis at low prices under the guidance of the government to meet the needs of tourists. Some countries, including those in Europe and Latin America have also begun to actively develop medical tourism by virtue of their national advantages. Medical tourism in China started late and developed slowly. As one of the pioneer cities in the development of medical tourism in China, Beijing has opened a medical tourism development model combining TCM hospitals and tourist attractions by virtue of the advantages of TCM [10]. For example, Guangdong Province has selected 19 demonstration bases for TCM cultural wellness tourism, with the aim of enhancing the brand of TCM cultural wellness tourism and preparing to try medical tourism. Sichuan is a major province of TCM resources in China, with advantages in TCM and ethnic medicine. It combines the two with the aim of creating a model for medical tourism development. Chengdu city is an important central city in the western part of the country, a national historical and cultural city, and a platform for cultural exchanges and economic cooperation in China. Chengdu city has abundant medical tourism-related resources but has not carried out medical tourism. This paper analyzes medical tourism in Chengdu city to promote the development of local medical tourism.

Today, with the comprehensive development and rapid development of the city’s smart medical tourism, as a municipality directly under the Central Government and a historic city, the dissemination of the city’s tourism image is of great significance to the city’s development. The development and protection of high-quality scenic spots have become the basic guarantee for the development of medical tourism in Chengdu city. The communication of the tourism image is the key to brand launch. Tourism resources are limited. From the marketing perspective, a city’s tourism resources can be considered commodities. The intrinsic value and the external brand value of a commodity together constitute the actual value of a commodity. From this point of view, the dissemination of the image of medical tourism in Chengdu city directly affects the sustainable development of medical tourism in Chengdu city and the rapid growth of the tourism economy. Effective image dissemination can play a role in promoting the development of tourism. Therefore, this paper constructs an urban smart medical system and positions the image of urban tourism based on big data analysis technology. It takes big data as the background and discusses urban smart medical care and urban tourism image dissemination under big data starting from the objective characteristics of big data, which will have unique research significance. It is hoped that references will be provided for developing urban smart medical undertakings and shaping the urban tourism image through these two aspects of research. The core research content is to build an urban smart medical system based on big data technology to ensure the system’s normal operation. In addition, the influence of Chengdu city will be enhanced to promote the integration and development of its cultural, medical, tourism, and other industries.

## 2. Materials and Methods

### 2.1. Introduction to Big Data Processing Tools

(1) Overview of Hadoop

Relational databases have been the best choice for data management for quite some time. However, in the era of big data, the diversification of data types and analysis requirements makes relational databases insufficient in many application scenarios. This section will provide a brief overview and summary of Hadoop, today’s most popular big data processing tool.

Hadoop is an open-source distributed systems infrastructure developed by the Apache Software Foundation. The architecture consists of two core parts: the Hadoop Distributed File System (HDFS) and the distributed computing system with MapReduce (Google MapReduce’s open-source implementation) as the core. Hadoop provides users with a distributed storage and computing programming platform that masks the underlying details of the system. HDFS has the advantages of high fault tolerance, large throughput, and good scalability. It allows users to deploy Hadoop on inexpensive hardware to form a distributed file system. It is suitable for applications with very large data collections. In addition, the MapReduce distributed programming model allows users to develop parallel applications without knowing the details of the underlying implementation of the distributed system. In addition, HDFS relaxes POSIX’s requirements, allowing users to access data in the file system as streams. The advent of Hadoop has made it easy for users to organize server resources and build cheap and efficient distributed application platforms. Users fully use the parallel computing and distributed storage capabilities of computer clusters to complete the analysis and processing of big data.

(2) SOA-related technologies

① Web Services Description Language (WSDL)

WSDL is the network services description language. It is a description language that describes WebServices and how to access them. WSDL is an XML-based language. It clearly defines information such as service name, service interface, service location, and supported data types. WSDL documents include four core elements, portType, message, types, and binding. These elements also include child elements, such as operation and part. These elements complete a complete description of the Web Scrvice. The meanings of each element are shown in Table 1.

② Simple Object Access Protocol (SOAP)

SOAP is a data communication protocol based on XML. SOAP is mainly used to access network services with cross-platform, cross-language, and simple and extensible features. The SOAP message contains elements, as shown in Table 2.

### 2.2. Design of Urban Smart Medical System Based on Big Data Analysis Technology

(1) Design Principles and Goals

The overall purpose of the system is:The system provides the infrastructure for big data in public facilities. Intel Quick-Path Interconnect technology is used to interconnect servers in various regions and realize the integration of computing, storage, and network resources through virtualization technology. The technology implements demand for high-level, dynamic scaling, on-demand allocation, and efficient cloud computing business models [11].The system provides a unified public service layer. “Public service layer” refers to a public business component designed for smart medical services. This component mainly includes database storage, image storage services, and message middleware [12].

Its purpose is to realize the integration and operation of the underlying business by simplifying the development of the underlying application, greatly reducing the running time, and improving the operating reliability of the system. This layer provides a service API to cloud computing service developers.

3.The system is used to build a sustainable development platform for smart medical services. The resource library of the smart medical system is based on all medical cloud computing application services [13]. This library contains services provided by the platform and third-party developers. Service resources’ scalability and sustainable evolution are achieved through the integrated management of services [14].

The overall structure of the smart medical platform is mainly to consider its openness, sustainable scalability, flexibility, and business looseness. As the key to the smart medical system, it must be continuously improved upon in the platform. This requires the platform to have an open interface to facilitate the development and access of third-party medical cloud services [15]. Healthcare cloud computing is a process of continuous and sustainable integration. It needs to be scalable in the system structure and can be continuously integrated into a business without major changes to the entire architecture. Every medical cloud computing business is composed of multiple different business interfaces. These businesses must implement a loose, scalable architecture.

When designing a specific business, the appropriate interface is used. In the upper-layer businesses, to reduce the user’s port service requirements, its business interface is roughened, reducing business requirements [16]. In low-level services, more detailed service interfaces are provided to ensure that the underlying business logic is sufficiently resilient [17].

(2) Architecture Design

Based on the analysis of the modeling and requirements of the smart medical platform, the logical structure of the system is carried out, as shown in Figure 1.

In Figure 1, the system architecture of the smart medical cloud service platform includes six layers: access, medical cloud service governance, medical cloud service resource, platform public service, platform runtime service, and infrastructure. In addition, the system also includes the overall security and monitoring system of the platform.

The first level is the underlying service. An underlying architecture service is a virtual machine with Docker as the core, which virtualizes many Linux servers and advanced storage systems into a huge, dynamically scalable, on-demand resource library. OpenStack is used to manage various resources in the virtual environment, including starting, storing images, and migrating instances. High-level applications’ security assurance is implemented based on the underlying architecture’s common computing functions. It put forward the optimal allocation method of public resources based on genetic algorithm and cloud computing, which takes the running time and cost of public resources as the optimization objectives, and put forward the adaptive function of population constraint [18].

The second level is the service in the execution phase of the platform. The platform execution stage provides a software stack for applications, including RabbitMQ, etc., necessary for asynchronous communication between services. Services separate the database access intermediary of business and data models. They have a Hadoop Distributed File System (HDFS), which supports multiple databases such as relational Oracle and non-relational HBase. In summary, the service level of the execution phase provides a fully context-dependent element for the program’s execution [19].

The third level is the public service level platform. Massive cloud computing services inevitably have a certain degree of repetition; for example, appointments and remote consultations involve relevant information from patients and doctors. The simultaneous development of the two will lead to system resource consumption, resulting in system management problems. Therefore, based on Service-Oriented Architecture (SOA), all public services are extracted to a common service level, and service decomposition is used to achieve the highest efficiency.

The fourth level is the resource of medical cloud computing. The center of medical resources is the core of the entire smart medical system. At the level of medical resources, all the application cases of medical cloud computing that publish and provide public information are carried out on this platform. These medical cloud computing services are not coupled with each other, and each independently solves service requirements through service groups. This business resource layer has strong scalability and can support the continuous integration of medical cloud computing services. Each medical cloud computing business is scalable and can adjust its resource utilization according to the customer load. The cloud architecture provided by the underlying architecture-level resource library can provide support for expanding and shrinking businesses.

The fifth level is the management of medical cloud computing. The management level of medical cloud computing undertakes the coordination and management of platform services. It contains four aspects: registry, service subscription, service routing, and load balancing. Service registration is a service catalog of the platform, and the address of the service needs to be maintained at the service registration center when the service is being accessed. Service registration can keep different services consistent in different places. At any Health Services Resource Center, once medical services are provided, they must be registered with the Health Services Registry.

The sixth level is access. The access level is an important gateway to the smart medical platform. It contains two important portals: 1. Smart access ports for different users. 2. The open interface on the platform used to access third-party platforms and integrate the Software Development Kit (SDK) into the system. The system targets different user groups, including four ports: patients, doctors, medical institutions, and managers. After the user logs in, the platform will display the corresponding port according to the group selected by the user during registration [20].

The seventh level is the security system. The security system covers the entire system and provides security guarantees for applications, data, and systems. Under limited license conditions, both users and application programming interfaces (APIs) on the platform must be accessed through an identical identity and permission. The secure transmission protocol keeps the user’s sensitive information confidential during transmission. The entire platform is used for routine security checks. Abnormal business is cleaned up. This operation ensures the availability of platform services.

The eighth level is the surveillance system. The monitoring system also covers the whole system. The monitoring system takes the monitored information as the input of load forecasting, outputs it to load forecasting, and then allocates business resources after forecasting. Log monitoring and collection, reconstruction, and mining can help find problems and provide value for the data mined. As an SOA-oriented architecture, its internal and external business must be monitored efficiently. In order to ensure the stability of the service, service load, abnormal service, call statistics, etc., are monitored.

The system uses the browser and server (BS) architecture patterns for developing software. The client uses a web-based user model, which mainly displays interfaces and data, and does not involve specific business logic. The server receives the client’s business requirements, processes them, and sends them to the client for display [21]. This work mainly uses Struct2, Spring, Hibernate, and other open-source architectures as the main research architecture. The hierarchical relationship of the implementation of the system is shown in Figure 2.

Figure 2 divides the architecture into presentation, service, and persistence layers. The presentation level is associated with the client and completes the display and interaction of data. Under the Struc2 architecture, the execution and the service layer are separated, and Action transmits all request data. A service hierarchy is a part that performs specific business logic. This layer receives data from Actions and provides Web Service requirements to other systems. After completing its logic, the service level saves the data in the database. The persistence level acts as an intermediate link between the service level and the database to solve the coupling problem between the business logic and data of the enterprise. A monolithic-level architecture is adopted for easy maintenance, expansion, and upgrade.

### 2.3. The Dissemination and Positioning of an Urban Tourism Image Based on Big Data Analysis Technology

The positioning of the city image is an important part of the city image, and its positioning is usually established fixedly, as the city’s central feature [22]. The positioning of the city is based on the characteristics of the city, based on the survey data, using 3C analysis (Corporation, Customer, Competitors), SWOT (strengths, weaknesses, opportunities, threats), and brand location map (Location analysis using competitors), etc. [23]. In the era of big data, the explosive development of information in big cities has brought new development opportunities and severe tests [24]. The problems that cities face are not only storing and processing data efficiently and inexpensively, but also querying and processing data quickly, flexibly, and stably [25]. From the massive data, this work can infer the opponent’s data and play a key role in its own position and strategy.

The city tourism image is positioned from four angles. 1. The target group should be identified: the travel brand [26]. The main and auxiliary crowd of the city should be caught. 2. Through the analysis of the basic identity of the audience, the audience is deeply excavated. 3. The core competitiveness of urban tourism brands is established, and the similarities and advantages of similar cities are compared with other similar cities. 4. The city image is divided into four aspects: target audience, value recognition, difference advantage, and image communication [27]. Figure 3 shows the positioning connotation of the urban tourism image.

(1) Construction of Big Data Platform

The data promotion activities of urban tourism are inseparable from big data support. Through the big data system, the original and dynamic data about the city obtained in publicity activities are processed and analyzed in real-time, thus constructing a complete information system [28]. The service platform of “big data” mainly includes public media, government-specific data collection, government Weibo, WeChat, client and forum, and the release of city information.

(2) The Positioning Target of an Audience Using Big Data Technology

Big data technology can be closer to users, and with the help of data and technology, users’ interests can be accurately integrated. The Audience Engine launched by Baidu Alliance in September 2012 is taken as an example; the Audience Engine takes hundreds of billions of points of user network behavior information as the basis of “big data”. The positioning technology is divided into six kinds of audience interest points, search keywords, browsing keywords, visiting pages, websites, and reselling. Demographics are divided into natural attributes, long-term interests, and short-term specific behaviors. Ultimately, the demographics are presented in all directions, enabling a full range of attribute insights and descriptions [29].

By using the characteristics of a specific city as a keyword, big data mining technology is used to collect and mine it and obtain relevant target audience and behavior data [30]. In a specific data mining stage, different keywords are selected according to different characteristics, such as Nanjing, Jinling, Jiangning, etc. In determining the keywords, the following basic guidelines should be followed. Take account of the historical and cultural context of the city, major events, activities, and resources, and use them as the basis [31,32].

When the data of urban residents is collected, it can be divided into basic and behavioral information of the audience. This information can help the experiment understand the actions of the audience, as shown in Figure 4:

The audience data obtained can be roughly divided into three types.

Basic information includes name, gender, age, education, income, occupation, address, contact information, etc.

Behavior information includes browsing history, duration, content, frequency, keywords and consumption time, content, frequency, requirements, etc.

Psychological information includes personality characteristics, original consumption intentions, expectations, post-consumption evaluation, consumption psychology, etc.

These three types of information are the main information in the database. In the era of big data, the idea of “audience-centric” information is widely sought after by people. From the audience’s perspective, the reasons and needs for tourists to participate in the city’s tourism image are analyzed to understand their goals and preferences. Data analysis and data mining are used to accurately classify and track network users to discover and understand the individual needs of users. The precise positioning process of urban tourism images based on big data is shown in Figure 5:

### 2.4. Research Methods and Data Sources

(1) Research methods and data sources for testing the effects of the urban smart medical system using big data

Research method: this part adopts a quantitative analysis method to study the testing effect of the urban smart medical system by big data. Quantitative analysis is a method to analyze quantitative characteristics, quantitative relations, and changes in social phenomena.

Data source: the test environment of the smart medical cloud service platform is based on three Linux servers with CentOS7 installed. Applications and databases are deployed by virtualizing the servers into several Docker containers. Table 3 shows the allocation of running containers in the test environment.

Nginx is deployed in the front end of a real test system in reverse proxy mode. Tomcat, which is used for system load balancing, deploys the complete code of the medical cloud service platform. The modules of various medical cloud services can be deployed separately in clusters. The test environment adopts a mixed deployment mode. MySQL, as the core storage component of the system, is deployed in a master–slave backup mode to ensure data storage security.

(2) Research methods and data sources of the positioning effect of urban tourism image communication based on big data analysis technology.

Research method: This part adopts the questionnaire method to investigate the effect of the positioning of urban tourism image communication based on big data analysis technology. The questionnaire survey method is the way to design questions and generate questionnaires in advance of the research, let the public fill in the questionnaire, and then conduct statistical analysis on the collected data and draw scientific conclusions. This paper will use the questionnaire survey method, mainly through the questionnaire survey conducted on local citizens and tourists in various representative scenic spots in Chengdu. The audience will answer the questions through the questionnaire, obtain a first-hand perception of the audience on Chengdu’s tourism image under the new media environment, and carry out the result analysis.

Data source: tourists fill out questionnaires and give full play to their personal and interpersonal networks. They ask their relatives and friends to help fill out questionnaires online and offline. A total of 420 questionnaires were filled out, eliminating incomplete and too different questionnaires. There are 400 valid questionnaires, with an effective rate of 95.24%, which is feasible. The basic information of the respondents is shown in Table 4.

## 3. Results and Discussion

### 3.1. Analysis of the Testing Effect of an Urban Smart Medical System Based on Big Data Analysis Technology

(1) Functional Test

The functions of the smart medical platform are mainly realized through the client interface of the smart medical system network to ensure the various functions required and the use of users. Since smart medical services serve different users, clinical trials are carried out on PC and tested on smart medical platforms. Here, a PC platform is taken as an example. The main problems on the mobile phone side are the performance of web pages and the usage habits of users. The main functions and services of the Smart Healthcare Cloud are tested, as shown in Figure 6.

(2) Performance Test

The performance test tests the medical system and uses tools to simulate the data to realize the pressure on the medical system. The first part is the load forecasting method discussed earlier to verify its accuracy in load forecasting and the system’s response under various stresses.

The method is used for the accuracy of load prediction at a given random load pressure, as shown in Figure 7. The *X*-axis represents time (minutes); the *Y*-axis is the pressure index. Queries Per Second (QPS) is the stress index in the test. Within 30 min, a random number of requests are transmitted to the system. The algorithm expects the load and value. Trends are plotted every 5 min. Therefore, the relationship between the load of the model and the real load is consistent. Therefore, multiple indices are used as the basis for the load algorithm to meet the forecast for the business expansion process.

In Figure 7, the predicted load values (predicted query rate) of 0 min, 5 min, 10 min, 15 min, 20 min, 25 min, and 30 min are 250, 1100, 875, 550, 200, 800, and 225, respectively; the actual load values (actual query rate) are 180, 1000, 850, 600, 250, 700, and 250. The calculation shows that the Mean Absolute Percentage Error (MAPE) at 0 min, 5 min, 10 min, 15 min, 20 min, 25 min, and 30 min are 38.89%, 10.00%, 2.94%, 8.33%, 20.00%, 14.29%, and 10.00%, respectively. The conclusion shows that from 0 min to 30 min, the load value changes dynamically. The load value is the highest at 5 min, and the mean absolute percentage error is the smallest at 10 min.

The operation of the system under different pressures is shown in Figure 8. The system response time is used as the evaluation index, and the different pressures of the system in the three are given.

The operation of the system under different pressures is shown in Figure 8. The system response time is taken as the evaluation index, and the system’s response under three different pressures is given. In the test environment, the smart medical cloud service platform runs well under three different pressure conditions

(3) Application of the System

Based on the idea of “Internet +”, the intelligent medical system service platform integrates medical treatment and network cloud technology. The smart medical platform is layered according to the “oriented” architecture and realizes health management based on the “cloud” model. A smart medical platform is a public information service that enables users to use multiple terminals for medical care in a geographically unlimited range. Unlike the previous isolated medical service system, the smart medical system improves scalability and reliability by concentrating resources, sharing information, and being supported by big data analysis technology. Hospitals and doctors can save all medical information in the system. Under three different pressures, the smart medical system platform can effectively improve the allocation of medical resources and improve the quality of medical services in hospitals within a certain range [33].

### 3.2. Analysis of the Dissemination and Positioning Effect of an Urban Tourism Image Based on Big Data Analysis Technology

This work mainly uses intercepting tourists to fill in questionnaires at scenic spots and giving full play to their personal and interpersonal networks, asking relatives and friends to help fill out the questionnaires online and offline. A total of 400 questionnaires are filled out. The main findings are shown in Figure 9.

In Figure 9, 43.41% of people believe that the positioning of Chengdu city represents local characteristics and can stimulate tourists’ desire to travel. In addition, 33.76% of people think that the positioning of Chengdu city has certain characteristics and can attract some tourists. Only 11.58% of people think that Chengdu city’s positioning feels average and not brilliant, and 8.36% of the people think that the positioning of Chengdu city does not match the local characteristics. A total of 2.89% of the people said they do not understand the positioning of Chengdu city. The data shows that the positioning of Chengdu city has a high degree of recognition.

In Figure 10, 12.6% of the respondents are completely satisfied with the tourism image of Chengdu city; 49.52% are satisfied; 35.05% are “general”; 2.25% are not very satisfied; 0.32% are very dissatisfied. Data show that more than half of the public is satisfied with the tourism image of Chengdu city.

## 4. Conclusions

The city’s smart medical system is designed based on big data analysis. Chengdu city is an example, and the city tourism image communication based on big data analysis technology is positioned. The results show that: (1) as a public information service system, the smart medical cloud platform of the urban smart medical system supports users to obtain medical services based on various terminal devices without geographical restrictions. The smart medical cloud realizes service aggregation and data sharing compared to the traditional isolated medical service system. Cloud computing improves the scalability and reliability of the system. (2) Based on the discussion and analysis of existing cases of using big data technology and methodology to carry out communication activities, this study discusses how to use tools in the big data environment, extract information from data, make image communication activities accurate, reduce the cost, and improve the efficiency of city image communication [34,35].

However, the system still has shortcomings and needs to be further improved. This work makes two recommendations. 1. The related technologies of big data have been deeply analyzed, and the platform of big data has been realized from the technical level. Additionally, all medical schools are inspected. Various problems in implementing the application of the big data platform are identified. The architecture of the smart medical system is used as the basis for detailed technical construction. In order to realize the sharing of regional medical resources, more information provided by the hospital can provide more convenience for patients and doctors. 2. In the future, various media will be more widely publicized, and audience marketing will be explored in depth.

## Figures and Tables

**Figure 1 ijerph-19-15330-f001:**
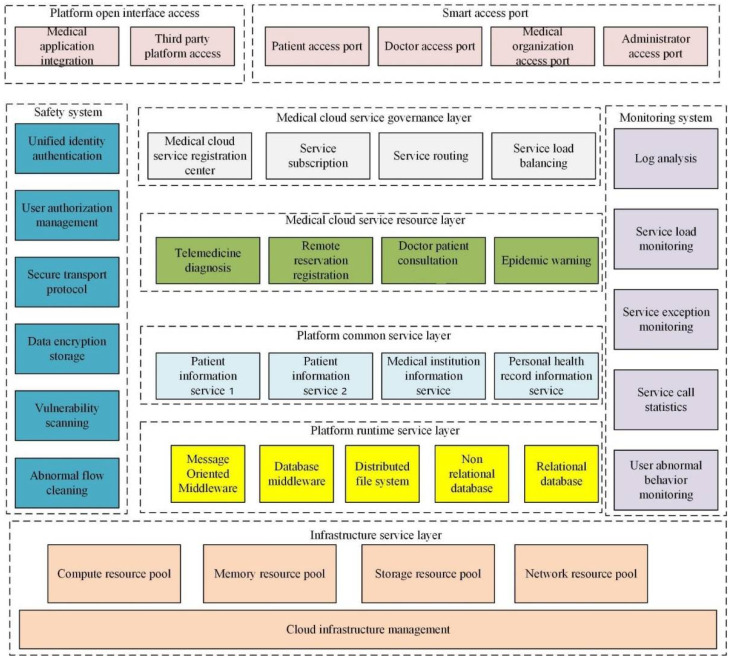
The architecture of the system platform of smart medical cloud service (drawn from research analysis after reviewing the data).

**Figure 2 ijerph-19-15330-f002:**
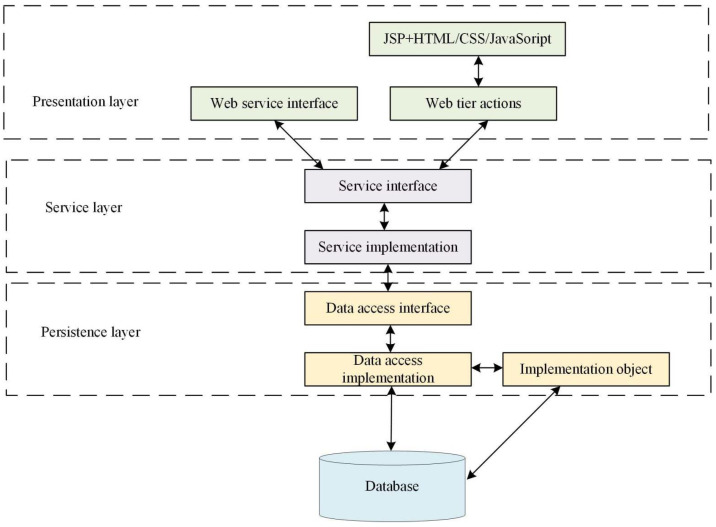
The structure of the system implementation hierarchy (drawn from research analysis after reviewing the data).

**Figure 3 ijerph-19-15330-f003:**
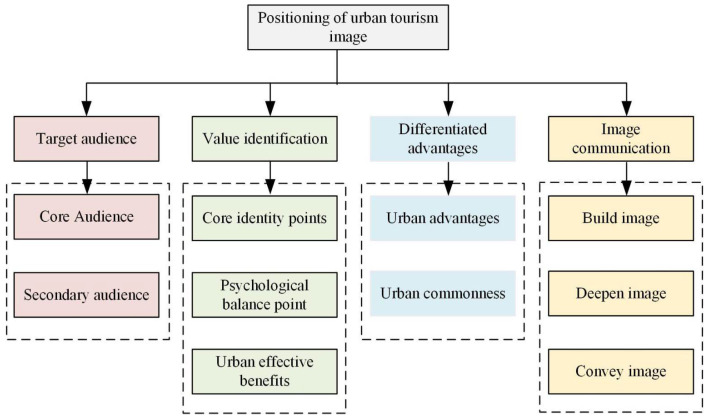
The positioning connotation of urban tourism image (drawn from research analysis after reviewing the data).

**Figure 4 ijerph-19-15330-f004:**
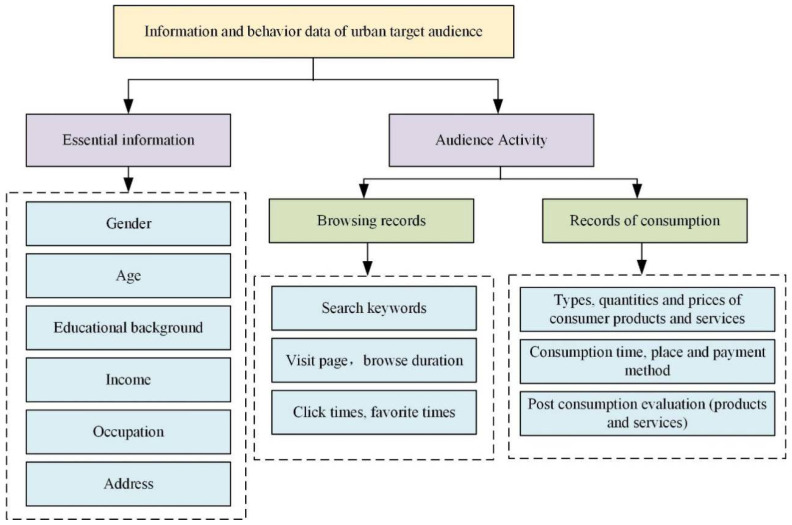
Information and behavior data of the city’s target audience (drawn from research analysis after reviewing the data).

**Figure 5 ijerph-19-15330-f005:**
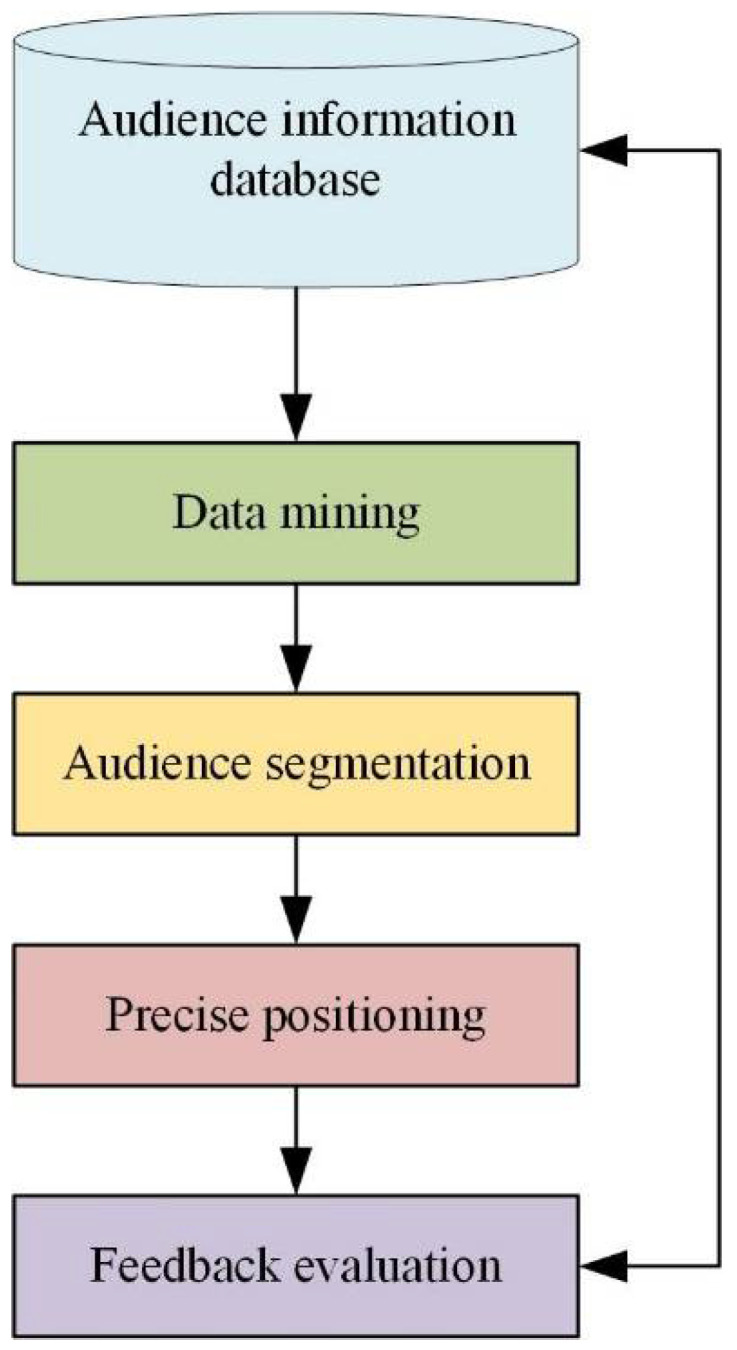
The process of precise positioning of big data (drawn from research analysis after reviewing the data).

**Figure 6 ijerph-19-15330-f006:**
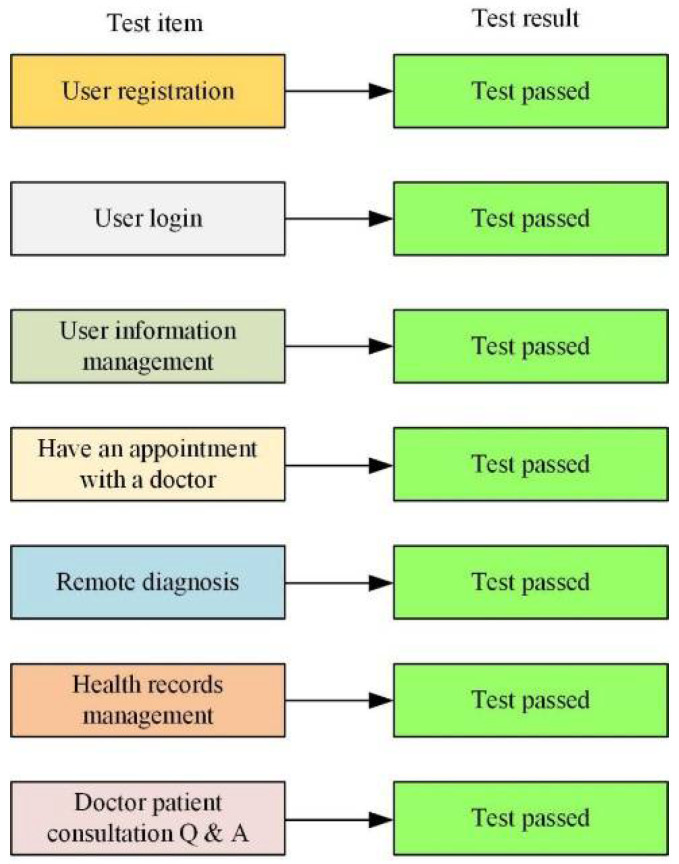
Testing of functions and services (drawn according to the test results after research and design combined with data).

**Figure 7 ijerph-19-15330-f007:**
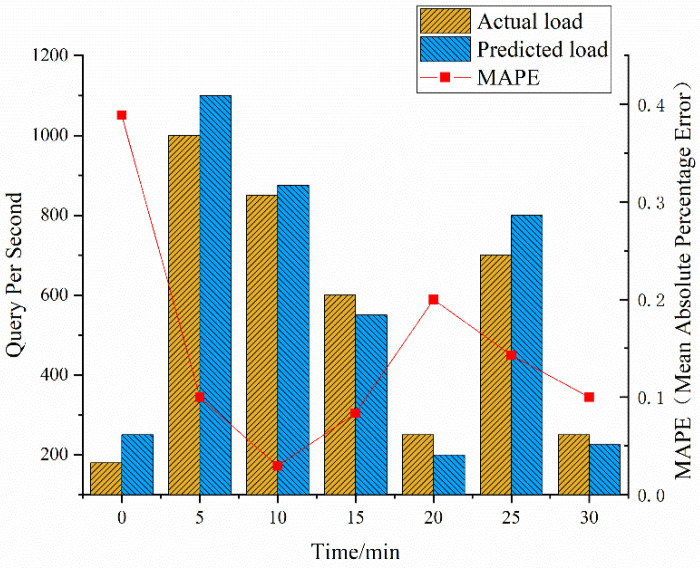
Test of the algorithm (drawn according to the test results after research and design combined with data).

**Figure 8 ijerph-19-15330-f008:**
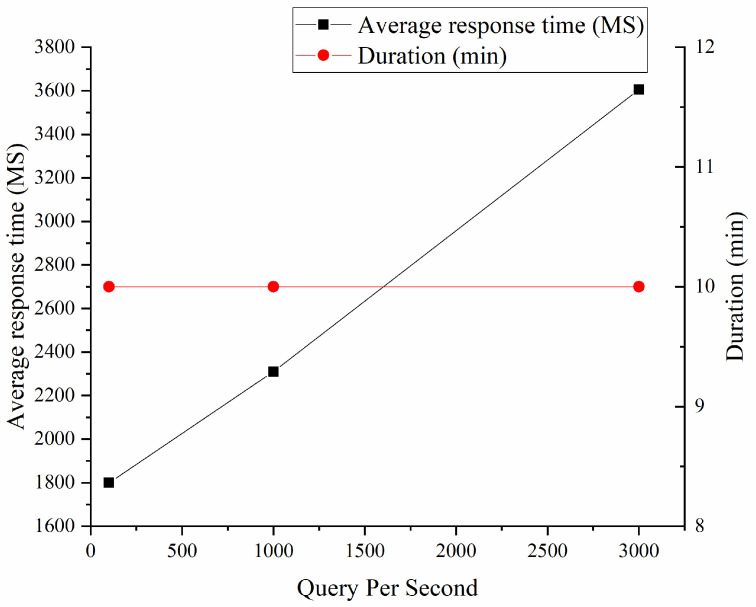
Test of stress (drawn according to the test results after research and design combined with data).

**Figure 9 ijerph-19-15330-f009:**
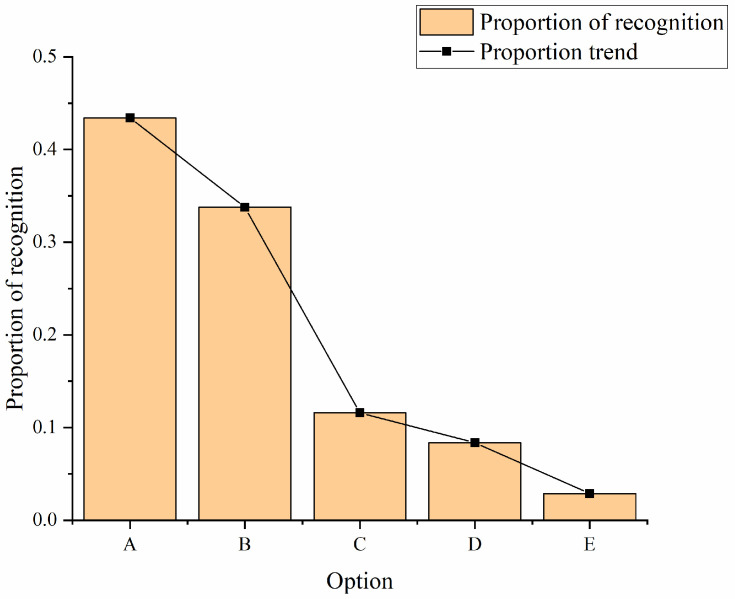
Respondents’ recognition of Chengdu city’s tourism image positioning (drawn according to the test results after research and design combined with data). (A: very representative; B: certain characteristics; C: Chengdu city’s positioning feels average and not brilliant; D: Chengdu city’s positioning does not match local characteristics; E: I don’t quite understand the positioning of Chengdu city).

**Figure 10 ijerph-19-15330-f010:**
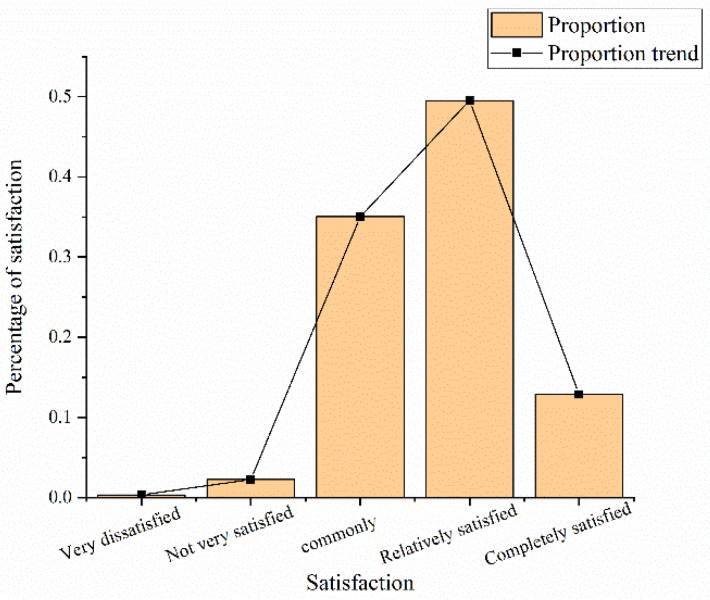
Survey respondents’ overall evaluation of the dissemination strategy of the urban tourism image based on big data analysis technology (drawn according to the test results after research and design combined with data).

**Table 1 ijerph-19-15330-t001:** Description of WSDL elements.

Element	Description
<portType>	Operations performed by the Web Service
<message>	Messages used by Web Service
<types>	The data type used by Web Service
<binding>	The communication protocol used by Web Service
<operation>	Abstractly describe the operations supported by the service
<part>	Parameters of the message
<port>	Defined as a single endpoint for a combination of binding and network address

The data was obtained by consulting the data.

**Table 2 ijerph-19-15330-t002:** SOAP elements.

Tool	Description
Envelope	Define the XML document as an SOA message
Header	Contains header information such as namespaces
Body	Contains call and response information
Fault	Provides error information during processing

The data was obtained by compiling the information after consulting the data.

**Table 3 ijerph-19-15330-t003:** Container deployment allocation.

Deployment Content	Illustration	Number of Container Allocations
Nginx	Load balancing server	1
Tomcat	Platform application server	3
MySQL	database	2

The table was obtained from research analysis after reviewing the data.

**Table 4 ijerph-19-15330-t004:** Basic information of respondents.

Item	Category	Number	Proportion
Gender	male	175	43.75%
	female	225	56.25%
Age	<30	45	11.25%
	31–40	77	19.25%
	41–50	172	43.00%
	51<	106	26.50%
Education background	junior college education	227	56.75%
	regular college education	134	33.50%
	graduate student education	39	9.75%

The table was obtained by compiling the information after consulting the data.

## Data Availability

The raw data supporting the conclusions of this article will be made available by the authors, without undue reservation.

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
