# Peer review of "The Dissemination Strategy of an Urban Smart Medical Tourism Image by Big Data Analysis Technology"

_ijerph, 2022, doi:10.3390/ijerph192215330_

Round 1

Reviewer 1 Report

The work presents a detailed description of the smart urban medical system based on big data analysis technology for the city of Beijing, and some analyzes (a load test and a satisfaction questionnaire) about this system. In general, I see a relevant contribution in the article, and in particular the details about the implementation of the health system management computer system is very interesting and thus already represents useful information for the IJERPH audience.

However, the empirical analyzes are quite summarized, and more details would be important to analyze the validity of the results shown. So I suggest a revision of the article, based on the incorporation of these points, as discussed in the points below.

1 – Incorporate a brief description of the analyses/methods used in the article in the abstract

2 - Define all acronyms used in the article

3 - The analysis performed in the Results and Discussion section needs to be discussed more carefully and in detail. The analysis deals with a particular aspect of the system, and is not a general analysis of the entire health management system. This analysis also needs to be included in the abstract and introduction of the article.

4 - The empirical analysis of section 4.2 also needs to be discussed in more detail, in particular the sample design, the response rate of the questionnaires, demographic aspects of the respondents, etc. In the current format, it is difficult to verify the validity of the sample design.

Author Response

1 – Incorporate a brief description of the analyses/methods used in the article in the abstract

Reply: Thank you for your advice. We have given a brief description of the research methodology used in the abstract.

2 - Define all acronyms used in the article

Reply: Thank you for your advice. Section 2.1 as well as abbreviations throughout the text have been explained in detail.

3 - The analysis performed in the Results and Discussion section needs to be discussed more carefully and in detail. The analysis deals with a particular aspect of the system, and is not a general analysis of the entire health management system. This analysis also needs to be included in the abstract and introduction of the article.

Reply: Thank you for your advice. Abstract, Introduction, and the discussion part of Section 4 analyze the specific aspects of the urban smart medical service system based on big data analysis technology and the urban tourism image communication positioning.

4 - The empirical analysis of section 4.2 also needs to be discussed in more detail, in particular the sample design, the response rate of the questionnaires, demographic aspects of the respondents, etc. In the current format, it is difficult to verify the validity of the sample design.

Reply: Thank you for your advice. In Section 2.3, the sample design of empirical analysis in Section 4.2, the response rate of the questionnaire, and the demographic statistics of the respondents are described in detail. The validity of the sample design has been verified.

Reviewer 2 Report

-       The paper must be structured more clearly.

-       The research question/hypothesis must clearly specify, as well as the research method.

-       The theoretical background must be improved.

-       Please insert the source to figures.

-      In the row 205 we find " the architecture's architecture is divided into presentation", maybe it would be better to use another word and not twice "architecture".

Author Response

  The paper must be structured more clearly.

Reply: Thank you for your advice. This paper has added research methods, data sources, and refined the summary, introduction, and conclusion to make the overall structure clearer.

-       The research question/hypothesis must clearly specify, as well as the research method.

Reply: Thank you for your advice. In this paper, the problems to be studied, methods and data sources have been determined in the introduction and Section 2.3.

-       The theoretical background must be improved.

Reply: Thank you for your advice. In the introduction, the theoretical background has been reorganized.

-       Please insert the source to figures.

Reply: Thank you for your advice. In Section 2.3, the research methods, and data sources of "Research methods and data sources for testing the effects of the urban smart medical system using big data" and "Research methods and data sources of positioning effect of urban tourism image communication based on big data analysis technology" have been described in detail.

-      In the row 205 we find " the architecture's architecture is divided into presentation", maybe it would be better to use another word and not twice "architecture".

Reply: Thank you for your advice. The text below Figure 2 has been modified.

Round 2

Reviewer 1 Report

The authors revise the manuscript according to all  my suggestions, and now the article is adequated for publication in IJERPH. 

Author Response

The authors revise the manuscript according to all my suggestions, and now the article is adequated for publication in IJERPH.

Reply: Thank you for your review and valuable comments.

Reviewer 2 Report

Dear author,

- Please follow the journal recommendation for structure the paper https://www.mdpi.com/journal/ijerph/instructions#preparation  

- The chapter 2. Method, I think the content is not only methods. My recommendation is to define the chapter as: Material and methods. 

- Please insert the source of the figures, example “Source: Authors’ computation based on……. or own research and edited). Example, The figure 1. It is your design, based on…?  Or this is from another work?

- The recommendation in the first version of paper was: “The theoretical background must be improved."

Your reply: Thank you for your advice. In the introduction, the theoretical background has been reorganized.”

The theoretical background must improve not only reorganize, the literature in the field of tourism, medical tourism, urban tourism and image should develop as starting point.

For example, the title contains Medical tourism, you write “the communication effect of the city's smart medical tourism image is analyzed” row 94, but you don't present anything about the Medical tourism, urban tourism, image in tourism, etc.

Author Response

- Please follow the journal recommendation for structure the paper https://www.mdpi.com/journal/ijerph/instructions#preparation  

Reply: The corresponding adjustment and optimization have been made in the appropriate part of the paper according to the requirements.

- The chapter 2. Method, I think the content is not only methods. My recommendation is to define the chapter as: Material and methods. 

Reply: The title of Chapter 2 has been changed to "Materials and Methods," as required. The introduction of the research tools has been supplemented.

- Please insert the source of the figures, example “Source: Authors’ computation based on……. or own research and edited). Example, The figure 1. It is your design, based on…?  Or this is from another work?

Reply: The data sources have been supplemented below the charts in the paper as required.

- The recommendation in the first version of paper was: “The theoretical background must be improved."

Your reply: Thank you for your advice. In the introduction, the theoretical background has been reorganized.”

The theoretical background must improve not only reorganize, the literature in the field of tourism, medical tourism, urban tourism and image should develop as starting point.

Reply: The introduction and analysis of medical tourism, urban tourism, and tourism image have been supplemented in the research background as required.

For example, the title contains Medical tourism, you write “the communication effect of the city's smart medical tourism image is analyzed” row 94, but you don't present anything about the Medical tourism, urban tourism, image in tourism, etc.

Reply: Relevant content introduction has been supplemented in the research background as required.